# Photodynamic Therapy in Primary Breast Cancer

**DOI:** 10.3390/jcm9020483

**Published:** 2020-02-10

**Authors:** Shramana M. Banerjee, Soha El-Sheikh, Anmol Malhotra, Charles A. Mosse, Sweta Parker, Norman R. Williams, Alexander J. MacRobert, Rifat Hamoudi, Stephen G. Bown, Mo R. S. Keshtgar

**Affiliations:** 1Breast Unit and Royal Free London NHS Foundation Trust, London NW3 2QG, UK; sohabobby@hotmail.com (S.E.-S.); anmolmalhotra@nhs.net (A.M.); 2Division of Surgery and Interventional Science, University College London, London W1W 7TY, UK; sandymosse@gmail.com (C.A.M.); sweta.parker@nhs.net (S.P.); norman.williams@ucl.ac.uk (N.R.W.); a.macrobert@ucl.ac.uk (A.J.M.); r.hamoudi@ucl.ac.uk (R.H.); s.bown@ucl.ac.uk (S.G.B.); m.keshtgar@ucl.ac.uk (M.R.S.K.); 3Department Cellular Pathology, Royal Free London NHS Foundation Trust, London NW3 2QG, UK; 4Department Radiology, Royal Free London NHS Foundation Trust, London NW3 2QG, UK; 5College of Medicine, University of Sharjah, Sharjah P.O. Box 27272, UAE

**Keywords:** breast cancer, photodynamic therapy, MRI, clinical study

## Abstract

Photodynamic therapy (PDT) is a technique for producing localized necrosis with light after prior administration of a photosensitizing agent. This study investigates the nature, safety, and efficacy of PDT for image-guided treatment of primary breast cancer. We performed a phase I/IIa dose escalation study in 12 female patients with a new diagnosis of invasive ductal breast cancer and scheduled to undergo mastectomy as a first treatment. The photosensitizer verteporfin (0.4 mg/kg) was administered intravenously followed by exposure to escalating light doses (20, 30, 40, 50 J; 3 patients per dose) delivered via a laser fiber positioned interstitially under ultrasound guidance. MRI (magnetic resonance imaging) scans were performed prior to and 4 days after PDT. Histological examination of the excised tissue was performed. PDT was well tolerated, with no adverse events. PDT effects were detected by MRI in 7 patients and histology in 8 patients, increasing in extent with the delivered light dose, with good correlation between the 2 modalities. Histologically, there were distinctive features of PDT necrosis, in contrast to spontaneous necrosis. Apoptosis was detected in adjacent normal tissue. Median follow-up of 50 months revealed no adverse effects and outcomes no worse than a comparable control population. This study confirms a potential role for PDT in the management of early breast cancer.

## 1. Introduction

Breast cancer is the most common cancer to affect women worldwide [1]. It has been the leading cause of cancer in women in Europe for many years, affecting 1 in 8 women in their lifetime although survival has been improving. These changes have been attributed to mammographic breast screening and improvements in surgical margins, chemotherapy, and the use of adjuvant hormonal therapies [2].

In spite of these successes, there remains a need for novel technologies in the treatment of all stages of breast cancer, not only to improve outcomes but also to allow more options for all patients, particularly those who are not eligible or responsive to standard management. A novel therapeutic modality that may offer an advantage to conventional therapies or even surgery in unfit patients is photodynamic therapy (PDT) [3,4].

PDT exerts its effects when light of a specific wavelength is used to trigger a photochemical reaction of a non-toxic photosensitizer (PS) previously injected systemically and given time to localize within the tumor and its vasculature. The reaction results in the production of superoxide anion radicals and reactive singlet oxygen molecules, the consequence of which is the initiation of cell death mechanisms. The nature of the biological effect may be at least partially localized to the tumor since neo-vascularization with “leaky capillaries” are tumor characteristics. This may result in a higher tissue concentration of the photosensitizer within the tumor. There is also a rapidly increasing body of evidence to show that PDT can stimulate strong immunological responses [5].

Photodynamic therapy is well established in dermatology for conditions such as non-melanoma skin cancers (basal cell carcinoma), Bowen’s disease, and pre-malignant conditions like actinic keratosis [6]. Non-dermatologic applications include all stages of head and neck cancer [7] and the non-neoplastic condition wet age-related macular degeneration of the retina (AMD) [8]. Image-guided interstitial PDT has been successfully used in advanced head and neck tumors [9] and clinical trials undertaken in several other solid tumors, including pancreatic and prostate cancers [10,11]. PDT for low-grade prostate cancer localized to the gland using the photosensitizer Tookad soluble™ has recently been approved for general use by the EMA (European Medicines Agency) [12]. Previous reports have described PDT amongst other options for the treatment of cutaneous metastases from breast cancer [13,14]. This report describes the first clinical study of PDT in the treatment of primary breast cancer.

## 2. Experimental Section

### 2.1. Study Design

This was a phase I/IIa study using PDT with the photosensitizer verteporfin to treat a small area in breast cancers scheduled for mastectomy, which was undertaken a few days after PDT. This was followed by a careful histological examination of the PDT treated area in the excised tissue. MRI scans were taken before PDT and immediately prior to surgery. The aims are to understand the nature of PDT effects on breast cancer and correlate the extent of effects, as measured by MRI and histology, to the PDT treatment parameters. The study was approved by the London-Hampstead Research Ethics Committee (REC reference: 09/H0720/123; EudraCT number: 2009–016276–74). All patients provided written informed consent.

### 2.2. Patient Selection

This was a single center, open label study of PDT in primary breast cancer. Women attending the breast clinic at the Royal Free Hospital, who had presented with a lump and been diagnosed to have primary breast cancer were recruited from April 2013 onwards. The treatment options, as recommended by the MDT (multidisciplinary team), were discussed with the patients. Patients who opted for surgery as their primary treatment and fitted the eligibility criteria were approached to participate in the study. Inclusion and exclusion criteria are shown in Table 1.

Patients interested in taking part in the trial were offered an opportunity to discuss the study with a member of the research team and given a copy of the patient information sheet (PIS). They had at least 24 h to consider their possible participation before being invited to provide written informed consent. For those agreeing to participate the following screening information was obtained:
Full medical history, including last menstrual period (LMP).Current medication.Pregnancy test if not postmenopausal or menstruating.Pre-treatment MRI scan of both breasts using dedicated breast coils, with and without contrast enhancement, if not done as a part of their initial diagnostic pathway.Routine blood investigations: blood count, urea and electrolytes, liver function tests.

If no significant contraindications were detected, the patient was scheduled for PDT.

### 2.3. Photodynamic Therapy

The patient was admitted on the day of treatment and routine observations made. No sedation was required for the procedure.

The photosensitizer, verteporfin (0.4 mg/kg), was dissolved in 5% dextrose and administered as a single intravenous infusion at 3 mL/min in a volume of 30 mL, followed by a 250 mL intravenous rapid flush of 5% dextrose. The same dose of photosensitizer was used for all patients and was the same as that used previously for patients with pancreatic cancer [10].

After an interval of 60 to 90 min, the most suitable access point on the skin, assessed by ultrasound, was infiltrated with local anesthetic and a 14 French gauge cannula (spring-loaded hollow needle covered by a sterile plastic sheath) was inserted 5 to 15 mm into the tumor under ultrasound guidance, avoiding areas of known spontaneous necrosis identified on the pre-treatment scans, where possible. The needle was removed and a 1-mm diameter laser fiber with a 1-cm diffuser tip inserted into the translucent plastic sheath, marked and fixed so the tip just reached the distal end of the sheath. Laser light (wavelength 690 nm, power 150 mW per cm of diffuser tip) could then be delivered to the chosen area of the tumor (Figure 1a,b). The total light dose delivered was 20 J to each of the first 3 patients, then 30 J to 3, 40 J to 3, and finally 50 J to 3.

Immediately after the light delivery, a second cannula was inserted alongside the first and a short (2 mm) titanium wire clip surrounded by a small amount of hydromarker gel was positioned using ultrasound to enable localization of the treatment site in the resected tissue. Ultrasound was used to ensure that the position of the clip was no more than 5 mm from the tip of the first cannula.

After the procedure, patients remained in subdued lighting in a side room on the ward for 24 h after photosensitizer administration, followed by re-adaptation to indirect sunlight for increasing periods while still an in-patient. Bright indoor light was permitted after 24 h and exposure to direct sunlight after 48 h, at which time they were discharged from the hospital. Blood investigations (FBC, U&E, and LFTs) were repeated 24 h after PDT.

Patients were readmitted a few days after PDT for a repeat MRI scan, clinical review, and for their scheduled mastectomy with axillary clearance. The MRI was performed as before, prone using a dedicated breast coil with contrast. The scans before and after PDT were reviewed to detect any spontaneous necrosis present before PDT and to estimate the volume of PDT induced necrosis in each patient by three-dimensional measurements on the scans. After discharge from hospital, follow-up continued in the breast clinic.

After surgery, the excised breast tissue was sent for macroscopic and histological assessment. The fresh specimen was sliced into 10–15-mm thick sections, which were examined macroscopically and photographed to identify the needle tract and adjacent marker clip site, and to document the extent of areas of necrosis detectable visually (Figure 1c). The sections were then fixed in formalin. After fixation, small blocks were taken from representative areas and sections prepared for histological examination using hematoxylin and eosin staining (H&E).

## 3. Results

Twenty-six patients were identified as eligible for the study, of whom 12 consented to participate, and they all completed the study (demographic summary in Table 2). All 12 had axillary nodal involvement at the time of diagnosis (median 3 lymph nodes, range 1–23), but none had visceral distant metastases. The remaining 14 patients had similar clinical criteria.

### 3.1. Quantification of PDT Effects

Nine patients had a mastectomy and axillary node clearance 4 days after PDT. Eight of the nine patients had their second MRI on the day of surgery. One (Patient 3), who initially agreed, was unable to have MRI scans due to claustrophobia. One (Patient 4) had surgery 7 days after PDT and 2 (Patients 10 and 11) after 11 days.

Pre- and post-PDT scans were compared to estimate the volume of the entire tumor, the volume of spontaneous necrosis present prior to PDT, if any, and the (additional) volume of necrosis associated with the PDT.

PDT effects were detected by MRI in seven patients. In six, the volume of PDT necrosis could be estimated and ranged from 78 to 8316 mm^3^. Pre-existing spontaneous necrosis was identified in four patients, including one (Patient 6) in which this overlapped with PDT necrosis, making it difficult to quantify the volume of necrosis related purely to PDT. Within the mastectomy specimen, the track of the needle used for fiber insertion was identified on all 12 patients and the marker was seen in 9, thereby guiding the pathological analysis. Examination revealed macroscopic necrosis related to PDT in 3 of the resected tumors, but microscopic necrosis was detected in 8 of the 12 patients, with a range of 65 to 8182 mm^3^. Histologically, the volume of PDT necrosis was calculated from tri-dimensional measurements made on tissue slices. There were four patients in whom no PDT effect was identified by MRI or histology. These results are summarized in Table 3, together with the depth of insertion of the laser fiber into the tumor. Figure 2 shows a comparison of the maximum diameter of the PDT effect, as determined by MRI and histology in areas free of spontaneous necrosis).

Patient 6 had her first MRI 270 days prior to PDT, so no MRI data was available immediately prior to PDT; however, the ultrasound scan used for inserting the laser fiber showed extensive necrosis extending into the planned PDT treatment area, so much of the light was delivered into necrotic tissue.

The histological findings are shown in Figure 3a–f. PDT necrosis was mostly confluent with clearly defined margins (Figure 3a), in contrast to more diffuse spontaneous necrosis identified on MRI prior to PDT (Figure 3b). In several cases, apoptosis was seen immediately adjacent to areas of necrosis (Figure 3c). Apoptosis has been previously reported after PDT and is associated with lower doses of photosensitizing drugs or light than necrosis [15]. The effect on normal breast glandular tissue was revealed in one patient (Patient 2) where instead of being placed deep into the tumor, the diffuser tip of the laser fiber was centered on the junction of the tumor and normal tissue. The normal tissue within the vicinity of the PDT fiber tract thus received a similar light dose to the cancer and showed apoptosis and mild acute inflammation, but no necrosis (Figure 3d). Healing of an area 11 days after PDT is shown in Figure 3e. Ducts expanded by DCIS showed zonal necrosis in response to PDT (Figure 3f). Vascular changes were evident within the necrotic area in the form of blood vessel occlusion, collapse, and extravascular RBC extravasation.

Three patients had evidence of hyaline necrosis in resected nodes, but there was no evidence of a PDT effect in any node.

### 3.2. Follow Up

There were no significant complications. After PDT, three patients required opiates in the first 24 h, the rest needing no more than paracetamol. All were asymptomatic, with no abnormal observations by 24 h. Full blood count, urea and electrolytes, and liver function tests were normal with no significant changes from pre-treatment results. There was no evidence of a PDT effect in the skin surrounding the site of needle insertion. All patients followed the advised regimen to avoid skin and eye photosensitivity and no-one had any problems. Following surgery, there were no adverse effects that could be correlated with the PDT and all patients made a routine post-operative recovery. Patients were followed up in the breast and medical oncology clinics after their post-operative review and received conventional treatment as recommended by the MDT.

The median follow-up was 50 months (range 19–58), during which time there was one distal metastasis (liver) after 3 years. All patients were alive at the most recent follow-up. In the control cohort comprising 14 patients who were eligible for the study but declined, there were 3 cases of distant metastases and one case of local recurrence.

## 4. Discussion

To our knowledge, this is the first clinical study of PDT in the treatment of primary breast cancer. Its main strength lies in the potential ability of PDT to target breast tumors that have shown no or minimal response to neoadjuvant therapy (NAT) [16]. The presence of residual disease after NAT indicates the existence of partial treatment resistance in the tumor, in which a minimally invasive local therapy such as PDT may play a role. PDT might also be able to play a role as an alternative to NAT prior to surgery in selected cases. Both offer the advantages of in vivo assessment of tumor response using MRI which may permit more effective use of conservative surgical procedures. PDT also has the potential to stimulate an immunological anti-tumor response [5].

PDT has been used previously to treat cutaneous metastases from breast cancers and has been associated with considerable discomfort. [13] In our study, by delivering the light energy directly into the bulk of the tumor via a laser fiber inserted through a needle positioned percutaneously, the procedure was well tolerated with minimal effect on the skin and surrounding tissue.

Similar to conventional neoadjuvant therapies, we have shown that performing MRI scans shortly before PDT and again a few days later, immediately prior to surgery, provides a potential tool to document the nature and extent of the changes related to PDT, compared to pathology as the gold standard. Precise MRI quantification of the extent of necrosis was often difficult, mainly due to the short time span between PDT treatment and tumor excision, before the full impact of the treatment became unequivocally visible on MRI. Furthermore, the presence of pre-existing areas of spontaneous necrosis was a confounding factor in some tumors, especially when the two types of necrosis overlapped. Longer intervals between PDT and follow-up MRI for small, well-defined cancers on pre-PDT MRI are likely to give a more accurate picture of the extent of the PDT effect.

Based on the use of a single laser fiber, there was a trend for the extent of necrosis to increase with the delivered light dose. At the initial dose of 20 J, necrosis was seen in 1 of 3 patients, but only on histology and of small volume. The extent of necrosis increased with increasing light dose. The absence of necrosis in Patient 11, who was treated with 50 J, was surprising. Considering that the needle tract was identified in the resected tumors in all cases on histology, it unlikely that this tumor or any other lesion was missed by the laser fiber or on histology. A rare technical problem such as an unrecognized laser fiber break cannot be excluded.

Because of the small number of cases included in this study, it was not possible to correlate the biological tumor characteristics, such as receptor status that predicts tumor susceptibility to other treatments, with PDT effects. For example, it has been shown in pancreatic cancer PDT that highly vascularized cancers are less responsive to PDT than less well vascularized lesions [17], but this observation was based on a small number of patients and it is not yet clear how important this aspect might be. Surrogate markers of vascular shutdown, including hypoxia-related protein expression combined with analysis of neovascularization, have been previously used in cholangiocarcinomas to prove the mechanism of cell death in PDT [18]. Similar tests could be undertaken on future patients similar to those in the present study as the morphological changes observed in breast adenocarcinomas so far support a similar mechanism. Upregulation of multidrug resistance proteins, especially P-glycoprotein 1, as well as the cytoprotective functions of some intracellular antioxidants like the glutathione system, catalase, lipoamide de-hydrogenase, and superoxide dismutase, which detoxify PDT-induced ROS, can result in PDT treatment resistance [19].

Comparable clinical studies of PDT with subsequent surgery within a few days have not been reported in other solid organs, so this study provides a unique insight into the histopathological changes that follow within a few days of treatment. The combination of vascular fibrinoid necrosis with extravasation of red blood cells, together with sharply demarcated necrosis with adjacent apoptosis, are the hallmark features of PDT-treated neoplasms. Within normal tissue, only those lobules within the immediate vicinity of the diffuser tip showed apoptosis. However, as apoptosis would not be expected in untreated normal tissue, this suggests a PDT effect on normal tissue.

This limited selectivity of PDT necrosis between cancers and the adjacent normal tissues in which the cancer arose has been known for many years [20]. However, normal glandular tissue in hollow organs heals largely by regeneration after PDT without significant loss of structure or function as connective tissues like collagen are largely unaffected, and so act as a scaffold to guide healing [21]. In laboratory studies on a larger, glandular organ, the canine prostate, PDT led to persistent glandular atrophy at 90 days, but with no disruption of the main stroma of the organ and no change in the ultimate size or shape of the gland [22]. Similar changes have been documented in the pancreas when an area of normal pancreas adjacent to a cancer was inadvertently treated with PDT [23]. Breast tissue has more adipose tissue than the pancreas or prostate and, in comparison, the glands are relatively sparsely packed. Thus, a PDT effect on normal breast tissue is much less likely to cause significant loss of normal tissue compared to surgery, making it acceptable to treat a rim of normal tissue around a breast cancer without significant risk of disfigurement.

An earlier study reported treating breast cancers with interstitial laser photocoagulation prior to surgery. The results correlating the extent of necrosis measured by MRI with that seen on histology were comparable to those in the present study; however, the biology of these thermal effects was quite different from those seen after PDT [24]. This is not unique to laser photocoagulation; studies that utilized any localized thermal treatment (e.g., laser, radiofrequency ablation, HIFU [25,26,27,28]) or cryotherapy showed no tumor selectivity and considerably more destruction of connective tissue, with healing by scarring and little glandular regeneration.

A major limitation of this study was slow recruitment. Breast cancer that requires mastectomy is emotionally affronting and the physical loss of the breast, as well as its implications for sexuality and body image, is difficult for most patients to come to terms with. Participation in a trial where “additional”, rather than alternative, and unproven treatment is proposed, which would not prevent the loss of the breast or guarantee any benefit to the future outcome, was not an attractive or worthwhile option for many of the patients who were eligible. We remain grateful to the 12 patients who did agree to participate. Our follow up data has so far (50 months) been reassuring and have confirmed that the women who participated in the trial had an outcome that was comparable to control cases, if not slightly better, with only one distant metastasis in the treated group and three distant metastases and one local recurrence in the control group. The difference in outcome was not statistically significant due to the sample size.

## 5. Conclusions

In summary, similar to some other types of solid tumors, this study has shown that PDT under image guidance is a promising, safe, and minimally invasive treatment for primary breast cancer that is reasonably predictable with minimal side effects on normal tissue compared to other local therapies. We consider that future studies should focus first on the best way to deliver PDT (e.g., drug dose and targeting, light dose, multiple fibers). Much current PDT research focuses on looking for ways of increasing the selectivity of PDT between tumors and their organ of origin, such as linking photosensitizers to tumor-specific antibodies, the use of nano-carriers for photosensitizing drug delivery, the latter particularly in multidrug-resistant breast cancer, where the photodynamic effect may play a role in bypassing and inhibiting escape pathways [29,30,31]. However, we see the best potential of PDT in breast cancer in the multi-therapy setting in patients with a poor or incomplete response to NAT or in cases where NAT and surgery are not possible. There may be circumstances in which PDT could be an acceptable alternative to surgery.

The beauty of PDT is that it is repeatable. If part of a tumor of known and limited extent is shown on MRI to have been incompletely treated, it is much simpler to repeat PDT than to repeat surgery, chemotherapy, or radiotherapy.

## Figures and Tables

**Figure 1 jcm-09-00483-f001:**
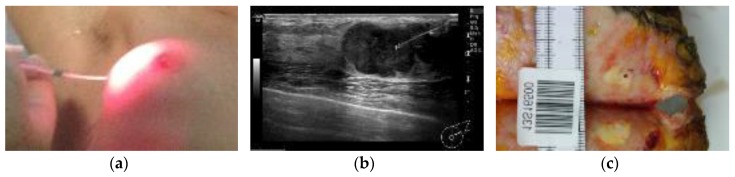
(**a**) Laser fiber inserted into the cancer for light delivery; (**b**) Ultrasound image of laser fiber in cancer; (**c**) Macroscopic appearance of the fiber track (the dark spot) in a homogeneous, pale area of PDT (Photodynamic Therapy) induced necrosis in the resected tissue cut perpendicular to the needle track. The lower homogeneous, pale area is the opposite side of the cut.

**Figure 2 jcm-09-00483-f002:**
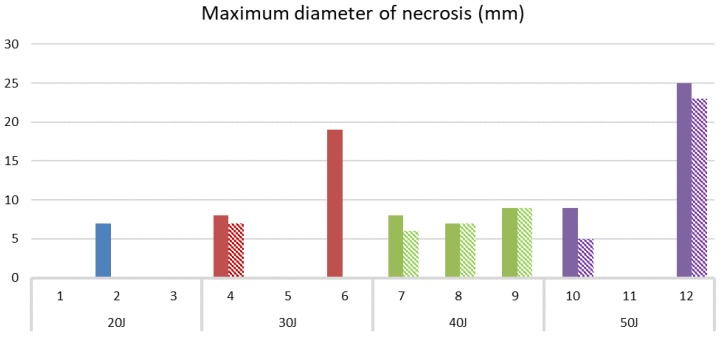
Maximum diameter of PDT necrosis in tumor in areas free of spontaneous necrosis prior to PDT as estimated by MRI (hatched bars) and histology (plain bars) for light doses from 20–50 J. On MRI, no PDT effect could be detected in Patient 2, and in Patient 6, the PDT effect could not be quantified on MRI as it was surrounded by spontaneous necrosis. Patient 10 had her surgery 11 days after PDT, by which time some healing had taken place. All patients had their repeat MRI 4 days after PDT.

**Figure 3 jcm-09-00483-f003:**
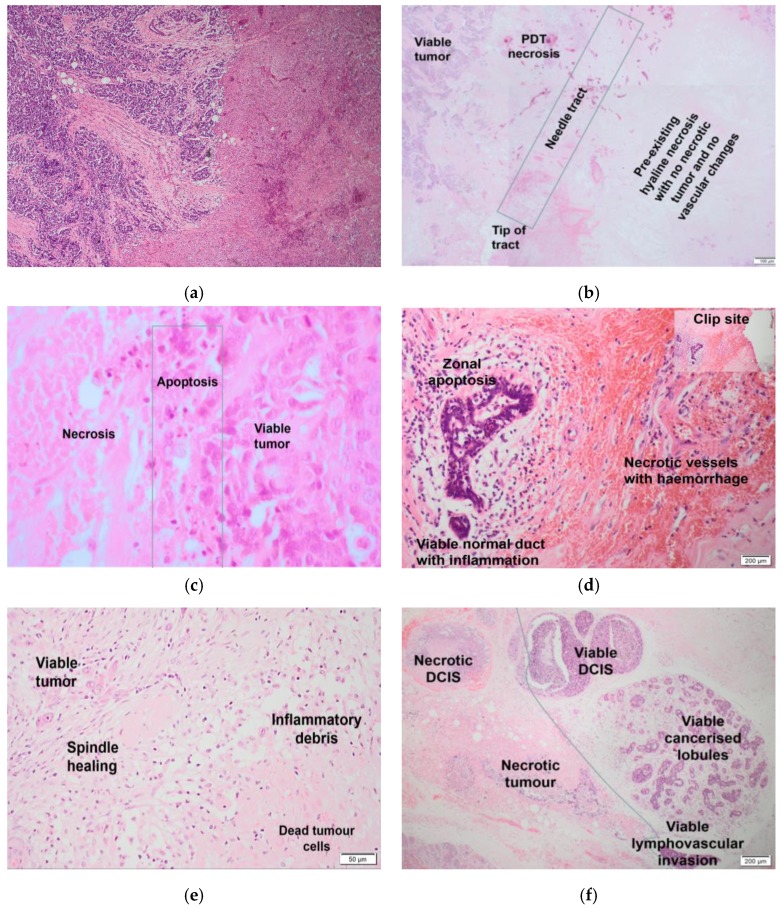
(**a**) Patient 12, 4 days after PDT. Low power view showing a sharply defined interface between viable (on the left) and PDT necrosed (on the right) tumor. There is congestion of blood vessels with extravasation of red cells in the necrosed area with some viable fat cells close to the adjacent viable tumor (magnification ×200). (**b**) Patient 6, 4 days after PDT. Composite images of section of tumor around fiber track showing acute coagulative PDT-induced necrosis on the left and hyaline necrosis/degeneration, suggesting subacute ischemic necrosis and identified on imaging prior to PDT on the right. (**c**) Patient 4, 7 days after PDT. High power magnification (×400) showing the frequently seen transition from viable tumor (on the right) to apoptosis (center) to necrotic tumor (on the left). (**d**) Patient 2, 4 days after PDT. On the left, there is viable normal tissue with inflammation and apoptosis. On the right, there is stroma and blood vessels, which are normal breast structures (but not glands), indicating a PDT effect in normal tissue. There is no tumor in this section. Proximity to the clip site is highlighted in the inset picture (top right) to show that this effect is definitely related to PDT and not incidental/far away. (**e**) Patient 10, 11 days after PDT. Healing in PDT treated tumor. Upper left: viable tumor cells entrapped within an area of plump fibroblasts, with macrophages and other inflammatory cells clearing debris from the necrotic tumor lower right. The small vessel in the center remains occluded by fibrin. (**f**) Patient 7, 4 days after PDT. The sharp interface between PDT-induced necrosis affecting DCIS (top left) and invasive tumor (bottom left), and viable DCIS (right) including the neoplastic cells populating a lobule below, is marked. A viable tumor embolus is seen within a small vascular channel (bottom right).

**Table 1 jcm-09-00483-t001:** Patient selection criteria.

Inclusion Criteria	Exclusion Criteria
1. Women age 30 years or over	1. Ductal carcinoma in situ (DCIS) without invasive carcinoma
2. Confirmed invasive ductal carcinoma (IDC)	2. Invasive lobular carcinoma
3. Unifocal tumor or unifocal site deemed suitable for PDT (Photodynamic Therapy) in multifocal invasive ductal carcinoma in a single breast	3. Current participation in any other trial of experimental medicine, or on current endocrine medication or neo-adjuvant therapy
4. Scheduled for surgery + axillary staging as primary treatment	4. Known metastatic disease
5. Negative pregnancy test within 7 days of registration for trial	5. Pregnancy and lactation
6. Willing to use contraception from date of consent until 6 weeks after completion of treatment	6. Severe cardiovascular or other systemic disease
7. Not breastfeeding	7. Known porphyria or sensitivity to photosensitizers
8. Capable of giving written informed consent	8. Any mental disorder making reliable informed consent impossible

**Table 2 jcm-09-00483-t002:** Characteristics of the PDT treatment cohort.

Characteristics	PDT Cohort
Total	12
Median age	49 (30–79)
Menopausal status	3 post-menopausal; 9 pre-menopausal
Estrogen (ER), Progesterone (PR) and Her 2 Receptor status: Group 1: ER + ve, PR + ve, Her 2 − ve	10
Group 2: ER − ve, PR − ve, Her 2 − ve	1
Group 3: ER + ve, PR + ve, Her 2 + ve	1
Tumor size	T2 = 4; T3 or greater = 8
Grade	G2 = 5, G3 = 7
Nodal status at mastectomy	All patients had positive nodes
Distant metastases at presentation	None
Primary treatment (after PDT)	Mastectomy
Adjuvant radiotherapy after PDT and mastectomy	12
Adjuvant chemotherapy after PDT and mastectomy	12
Adjuvant endocrine therapy after PDT and mastectomy	9

**Table 3 jcm-09-00483-t003:** Tumor assessment and treatment response.

Pt	Age	MRI Total Tumor Vol (mm^3^)	Pre-PDT Necrosis (mm^3^)	PDT Dose (J)	PDT Necrosis on MRI (mm^3^)	PDT Necrosis on Histology (mm^3^)	Depth of Needle Tip (mm)
1	79	21,800	170	20	0	0	n/a
2	45	91,100	1400	20	0	180	5
3	53	n/a	n/a	20	n/a	0	n/a
4	57	32,800	<10	30	180	65	6
5	36	198,000	0	30	0	0	n/a
6	49	433,000	2600	30	#	5576	12
7	30	135,000	0	40	78	270	10
8	48	235,000	0	40	180	180	13
9	47	3700	0	40	109	380	n/a
10	57	27,800	0	50	253	380	10
11	54	27,500	0	50	0	0	8
12	49	51,400	0	50	8316	8182	11

# Too much overlap between areas of spontaneous and PDT necrosis to estimate the volume of PDT effect on MRI (Magnetic Resonance Imaging). n/a, data not available as patient declined MRI due to claustrophobia. The maximum diameter of necrosis attributable just to PDT for all cases is shown in Figure 2.

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
