# Peer review of "Photodynamic Therapy in Primary Breast Cancer"

_jcm, 2020, doi:10.3390/jcm9020483_

Round 1

Reviewer 1 Report

Reviewer report jcm-700744

The authors present a light dose escalation PDT study for primary breast cancer, where outcomes were correlated to MRI and histology of the tumors. The advantage of the study is that the PDT session was succeeded by a mastectomy, allowing analysis of tumoral vascularization (i.e., delivery routes for the photosensitizer), photosensitizer retention, and PDT-mediated anti-cancer effects in the target tissue. The use of an interstitially position isotropic diffuser in patients is sheerly brilliant because many clinicians refrain from invasive procedure fearing needle track seeding. Here the authors showed that this scare is unnecessary. The study is adequately described, logically presented, and is of importance to patients, clinicians, and the research community. It perfectly fits in the scope of the journal.

Authors should note that page and line number(s) are presented as 2/45 (page 2, line 45) in the reviewer report.

GENERAL / MAJOR

1.    This study presented a rare opportunity to test post-PDT histological effects that are pertinent to therapeutic outcome. Firstly, the authors have access to formalin-fixed tissue samples, with which they could have assessed PDT-induced vascular shutdown (e.g., PMID 15930363 and Laser Physics Letters 2012; 10(2):023001). No one has really demonstrated this effect in patients. Vascular shutdown is key to inducing intratumoral hypoxia and metabolic catastrophe in parenchymal cells, which culminate in tumor cell death. PDT-induced vascular occlusion in hence instrumental in therapeutic efficacy. The staining can be routinely performed by histologists in academic hospitals, see e.g., https://www.ncbi.nlm.nih.gov/pmc/articles/PMC4823110/. I strongly recommend that the authors re-visit the histological sections and perform this analysis, which can be easily performed on the formalin-fixed slices. Secondly, the authors have alluded to the anti-tumor immune response that is induced by PDT. It would greatly benefit the field if this phenomenon would finally receive robust clinical validation, which can be performed using the histological slides from this clinical cohort. What type of immune cells are recruited, where are they localized, etc. are some of the key questions that could be answered.

2.    It would be useful to juxtapose the recurrence-free survival and metastasis rates to the statistics of patients who had been treated surgically (mastectomy) and adjuvant therapy, taken for example retrospectively from a patient database. This would give us an indication, albeit needing more clinical validation, of the potential advantage of PDT + surgery + adjuvant therapy over surgery + adjuvant therapy. A reference to existing literature featuring these statistics would also suffice.

SPECIFIC / MINOR

1.    2/49: it is advised that the segment “PDT can stimulate strong immunological responses” is rephrased to “PDT induces an anti-tumor immune response that is instrumental in therapeutic efficacy and clinical outcome.”

2.    3/108: better resolution images should be provided, if available, and copyedited into the manuscript. In the reviewer version the images are blurred due to poor resolution. Please annotate panel C in Figure 1, pointing to the site where the fiber was positioned.

3.    4/121: write out every acronym at first mention.

4.    4/135-140: How did the authors exclude chemoattracted and accumulated leukocytes from the Ki-67 quantitative analysis of proliferating tumor cells? See for clarification https://www.ncbi.nlm.nih.gov/pmc/articles/PMC3534316/. PDT-induced necrosis and apoptosis is expected to lead to a profound immune response characterized by the migration of various types of immune cells into the afflicted tissue volume. This phenomenon is clearly visible in Figure 3d.

5.    4/146: Table 1 - please indicate when the adjuvant therapies were administered. I presume this was after mastectomy as you indicate “after treatment,” but PDT is also a treatment. So better to state “…after primary treatment.”

6.    5/156-157: please provide a quantitative measure for V and CR. In other words, what does a vascularity score of 1 signify numerically (e.g., 15% of tumor tissue vascularized). Approximations are fine as long as readers can have a more concrete idea. These values should be reported in the Methods section on MRI image analysis.

7.    6/177: MR 270 should read MRI 270.

8.    6/181: Figure resolution is poor.

9.    7/218: Figure resolution is poor, as a result of which it is very difficult to assess the histology. Based on what can be seen, I recommend that the authors enhance the images in Photoshop to at least make the hues from eosin and hematoxylin comparable between images and standardize the histology presentation.

Author Response

The authors present a light dose escalation PDT study for primary breast cancer, where outcomes were correlated to MRI and histology of the tumors. The advantage of the study is that the PDT session was succeeded by a mastectomy, allowing analysis of tumoral vascularization (i.e., delivery routes for the photosensitizer), photosensitizer retention, and PDT-mediated anti-cancer effects in the target tissue. The use of an interstitially position isotropic diffuser in patients is sheerly brilliant because many clinicians refrain from invasive procedure fearing needle track seeding. Here the authors showed that this scare is unnecessary. The study is adequately described, logically presented, and is of importance to patients, clinicians, and the research community. It perfectly fits in the scope of the journal.

 Thank you for these positive remarks

Authors should note that page and line number(s) are presented as 2/45 (page 2, line 45) in the reviewer report.

GENERAL / MAJOR

This study presented a rare opportunity to test post-PDT histological effects that are pertinent to therapeutic outcome. Firstly, the authors have access to formalin-fixed tissue samples, with which they could have assessed PDT-induced vascular shutdown (e.g., PMID 15930363 and Laser Physics Letters 2012; 10(2):023001). No one has really demonstrated this effect in patients. Vascular shutdown is key to inducing intratumoral hypoxia and metabolic catastrophe in parenchymal cells, which culminate in tumor cell death. PDT-induced vascular occlusion in hence instrumental in therapeutic efficacy. The staining can be routinely performed by histologists in academic hospitals, see e.g., https://www.ncbi.nlm.nih.gov/pmc/articles/PMC4823110/. I strongly recommend that the authors re-visit the histological sections and perform this analysis, which can be easily performed on the formalin-fixed slices. Secondly, the authors have alluded to the anti-tumor immune response that is induced by PDT. It would greatly benefit the field if this phenomenon would finally receive robust clinical validation, which can be performed using the histological slides from this clinical cohort. What type of immune cells are recruited, where are they localized, etc. are some of the key questions that could be answered.

We thank the reviewer for these suggestions, which are highly appropriate, but regret to say that it is not feasible to undertake these additional studies now. Due to the prolonged illness and ultimate sad passing away of the Principle Investigator and key clinical supervisor, Prof Keshtgar, there has been a delay of several years in writing up this work and most of the team now have major commitments to other current projects or have moved on. The results we have on vascularity are only qualitative and only apply to the untreated tumours as seen on MRI. These do not show any notable results and so these data have been removed. Certainly, the vascular and immunological effects of PDT are of great interest, but detailed study of them in the breast will have to await future studies. We have added the following sentence to the discussion: Surrogate markers of vascular shut down including hypoxia-related protein expression, combined with analysis of neovascularisation have been previously used in cholangiocarcinomas to prove the mechanism of cell death in PDT (ref). Similar tests could be undertaken on future patients similar to those in the present study as the morphological changes observed in breast adenocarcinomas so far, support a similar mechanism. 

It would be useful to juxtapose the recurrence-free survival and metastasis rates to the statistics of patients who had been treated surgically (mastectomy) and adjuvant therapy, taken for example retrospectively from a patient database. This would give us an indication, albeit needing more clinical validation, of the potential advantage of PDT + surgery + adjuvant therapy over surgery + adjuvant therapy. A reference to existing literature featuring these statistics would also suffice.

We agree that in the long run, this would be a good way to proceed, but feel that the small number of patients in the present study would not justify such a comparison. Also, no attempt was made to treat the full volume of these cancers. The aim was only to treat a small area of cancer in an area adjacent to normal tissue and any comment on the value of PDT in extending recurrence free survival would not be of value unless an attempt had been made to treat the whole tumour volume.

 SPECIFIC / MINOR 

2/49: it is advised that the segment “PDT can stimulate strong immunological responses” is rephrased to “PDT induces an anti-tumor immune response that is instrumental in therapeutic efficacy and clinical outcome.”

We do not feel strongly about this, but the editors have asked us specifically to keep to our original terminology, so we would prefer to leave this expression as it is.

3/108: better resolution images should be provided, if available, and copyedited into the manuscript. In the reviewer version the images are blurred due to poor resolution. Please annotate panel C in Figure 1, pointing to the site where the fiber was positioned.

We are very conscious of the problem with fig 1a, but unfortunately, it’s the best image we have of light delivery. The images are all several years old. The legend for fig 1c has been considerably improved to read:

Fig 1c: Macroscopic appearance of fiber track (dark spot) in pale, homogeneous area of PDT induced necrosis on the surface of the resected mastectomy specimen cut perpendicular to the line of the fibre track. The lower pale area is on the opposite side of this cut.

4/121: write out every acronym at first mention.

This should have been done before and has now been done

4/135-140: How did the authors exclude chemoattracted and accumulated leukocytes from the Ki-67 quantitative analysis of proliferating tumor cells? See for clarification https://www.ncbi.nlm.nih.gov/pmc/articles/PMC3534316/. PDT-induced necrosis and apoptosis is expected to lead to a profound immune response characterized by the migration of various types of immune cells into the afflicted tissue volume. This phenomenon is clearly visible in Figure 3d.

Thank you for this important observation on fig 3d. However, as for the comments on vascularity, it is not currently realistic for us to go into any further detail at this point in time. We do not believe the Ki67 results have any effect on the main purpose of this paper, so we think the best approach is to remove all mention of Ki67 from this paper. Again, this is an important area for future research.

4/146: Table 1 - please indicate when the adjuvant therapies were administered. I presume this was after mastectomy as you indicate “after treatment,” but PDT is also a treatment. So better to state “…after primary treatment.”       

9/375-9 No adjuvant therapies were given prior to PDT and mastectomy. That was the key factor determining which patients were invited to join the study, as indicated in table 1. After treatment has been changed to after PDT and mastectomy. We do not have data on what adjuvant therapies were administered and when. However, our only purpose on long term follow up in this phase I/IIa study was to ensure there was no evidence that PDT had done any harm to any patients. The numbers were very small, but we did this from data on the patients who fitted the inclusion factors for the trial comparing long term results in those who agreed to participate in the trial and those who declined. This data is summarised in one sentence on Page 9 lines 375-9

5/156-157: please provide a quantitative measure for V and CR. In other words, what does a vascularity score of 1 signify numerically (e.g., 15% of tumor tissue vascularized). Approximations are fine as long as readers can have a more concrete idea. These values should be reported in the Methods section on MRI image analysis.

We have to admit that these are purely qualitative results based on visual interpretation of MRI images (as was stated in the text). Nothing particularly valuable came from these data, so we feel these results are best removed

6/177: MR 270 should read MRI 270.

This has been corrected.                                                                          

6/181: Figure resolution is poor. 

Fig 2 We have improved the image by going back to the original

7/218: Fig 3. Figure resolution is poor, as a result of which it is very difficult to assess the histology. Based on what can be seen, I recommend that the authors enhance the images in Photoshop to at least make the hues from eosin and hematoxylin comparable between images and standardize the histology presentation.

Fig 3 We knew this was a problem. We have endeavoured to sharpen and enhance the pictures as much as possible without affecting their integrity. Adjusting the hues to standardise the histology sections within a single panel is not entirely possible when the images are from different cases/sections and showing different magnifications.

Reviewer 2 Report

JCM-700744
Photodynamic therapy in Primary Breast Cancer

This an excellent article by Banerjee and colleagues that describes promising results from a well-considered study, under the supervision of highly respected physician-scientists. Dr. Keshtgar had a well-earned reputation as a talented and compassionate physician who was deeply dedicated to his patients. This article serves as another exceptional piece of his legacy. Dr. Bown is a highly respected expert in clinical photodynamic therapy (PDT) who has provided critical guidance and perspective to numerous high-quality studies around the world. This study presents a thoughtful clinical framework, and perspective, that will guide the potential inclusion of PDT in the armamentarium to manage patients with primary breast cancer, particularly those with no or minimal response to NAT. The insights and results presented here are of high value to the PDT community. Pending a few minor critiques, the article is enthusiastically recommended for publication, with deep respect for Dr. Keshtgar and his legacy.

1.     Please clarify the appropriateness of referring to PDT as an "ablative therapy." This is done twice in the manuscript. The traditional definition of ablation is vaporization of tissue due to heating (100°C). As the authors know well, PDT is a non-thermal modality that induces cytotoxicity via triplet-state photochemistry. The phrase “ablative therapy” is, therefore, incongruous with the photochemical mechanisms that are inherent to PDT. Please explain or modify this text.

2.     Do the authors believe that vascularity should be included in guiding light dosimetry in primary breast cancer, as (mentioned by the authors) it is in pancreatic cancer?

3.     Based on published studies showing the ability of PDT to reverse resistance and synergize with conventional therapies, do the authors believe the priority for future studies is to evaluate targeting/nanotechnology approaches (as suggested) or to develop PDT-based combinations? Is one approach considered by the authors to benefit patients most by preventing loss of the breast and/or providing improved outcomes to overcome recruitment hurdles in the short term, and adoption in the longer term?

Author Response

This an excellent article by Banerjee and colleagues that describes promising results from a well-considered study, under the supervision of highly respected physician-scientists. Dr. Keshtgar had a well-earned reputation as a talented and compassionate physician who was deeply dedicated to his patients. This article serves as another exceptional piece of his legacy. Dr. Bown is a highly respected expert in clinical photodynamic therapy (PDT) who has provided critical guidance and perspective to numerous high-quality studies around the world. This study presents a thoughtful clinical framework, and perspective, that will guide the potential inclusion of PDT in the armamentarium to manage patients with primary breast cancer, particularly those with no or minimal response to NAT. The insights and results presented here are of high value to the PDT community. Pending a few minor critiques, the article is enthusiastically recommended for publication, with deep respect for Dr. Keshtgar and his legacy.

Thank you for these kind remarks

Please clarify the appropriateness of referring to PDT as an "ablative therapy." This is done twice in the manuscript. The traditional definition of ablation is vaporization of tissue due to heating (100°C). As the authors know well, PDT is a non-thermal modality that induces cytotoxicity via triplet-state photochemistry. The phrase “ablative therapy” is, therefore, incongruous with the photochemical mechanisms that are inherent to PDT. Please explain or modify this text.

We take your point. We think of ablation as local destruction of tissue, such as necrosis or apoptosis, but if you see it as vaporisation, then clearly, we have used the term inappropriately. We have revised the text to remove any reference to PDT ablation. However, we should like to mention that the word ablation is now in common use when referring to RFA (radiofrequency ablation) as in the treatment of dysplasia in Barrett’s oesophagus and in image guided treatment of lesions in the centre of solid organs like the liver, lung and breast (our ref 25, mentioned in line372, p9). In these situations, the tissue is coagulated rather than vaporised.

Do the authors believe that vascularity should be included in guiding light dosimetry in primary breast cancer, as (mentioned by the authors) it is in pancreatic cancer?

Possibly, sometime in the future. The paper on pancreatic PDT, referred to in this manuscript, is the first (to our knowledge) to suggest that vascularity could be included in guiding light dosimetetry. Much more data will be required to assess the value of this. A comment has been added to say that this observation was based on a small number of patients

Based on published studies showing the ability of PDT to reverse resistance and synergize with conventional therapies, do the authors believe the priority for future studies is to evaluate targeting/nanotechnology approaches (as suggested) or to develop PDT-based combinations? Is one approach considered by the authors to benefit patients most by preventing loss of the breast and/or providing improved outcomes to overcome recruitment hurdles in the short term, and adoption in the longer term?

We believe that the first challenge should be to optimise PDT on its own, with or without targeting/nano technology. It is potentially so simple, safe and relatively cheap (main cost is price of photosensitiser) compared with surgery, chemotherapy and radiotherapy, for localised cancers. Then, combinations of PDT with other therapeutic options can be explored. Comments to this effect have been added to the conclusion.

Reviewer 3 Report

This is a very nice study of PDT for breast cancer.  Minor concern exists regarding the apparent lack of correlation between necrosis and dose.  Is this due to heterogeneity of response, heterogeneity of PDT dose (either optical properties or PS distribution)?  Any data that were collected that could be used to analyze for optical properties or PS distribution would be helpful in this regard.  If such data are not available, it would be good to expand the discussion of this important issue as it has strong implications for the success of a multifiber protocol with similar design, which is presumably the next step in the evolution of interstitial Breast PDT.

Author Response

This is a very nice study of PDT for breast cancer.  Minor concern exists regarding the apparent lack of correlation between necrosis and dose.  Is this due to heterogeneity of response, heterogeneity of PDT dose (either optical properties or PS distribution)?  Any data that were collected that could be used to analyze for optical properties or PS distribution would be helpful in this regard.  If such data are not available, it would be good to expand the discussion of this important issue as it has strong implications for the success of a multifiber protocol with similar design, which is presumably the next step in the evolution of interstitial Breast PDT.

Thank you for these kind comments. This was a very small study and there were various practical factors that undoubtedly meant that not all patients were treated in exactly the same way. There were also likely heterogeneities in the nature of the treated tissue, light distribution, concentration of photosensitiser etc as well as difficulties distinguishing between spontaneous and PDT induced necrosis on the MRI scans. All these could explain discrepancies between extent of necrosis and drug/light doses. Our work on PDT for pancreatic cancer has explored some of these factors, as in the referenced papers, but these aspects will need to be considered later for breast cancer.

The beauty of PDT, when better defined for the breast, is that if part of a treated tumour is missed, this should be detectable on MRI and treatment can be repeated. It is much more difficult and fraught with complications to repeat surgery, chemotherapy or radiotherapy, all of which can be associated with cumulative toxicity, which is not the case with PDT. A comment along these lines has been added to the conclusion.

Round 2

Reviewer 1 Report

Thanks for explaining the situation; my condolences to you for losing your colleague. I understand and congratulate you on the manuscript's acceptance. It is important work.